# Histological, Immunohistochemical and Antioxidant Analysis of Skin Wound Healing Influenced by the Topical Application of Brazilian Red Propolis

**DOI:** 10.3390/antiox11112188

**Published:** 2022-11-04

**Authors:** Mariana Conceição, Lucas Fernando Sérgio Gushiken, Jennyfer Andrea Aldana-Mejía, Matheus Hikaru Tanimoto, Marcos Vital de Sá Ferreira, Andreia Cristina Miranda Alves, Marina Naomi Miyashita, Jairo Kenupp Bastos, Fernando Pereira Beserra, Cláudia Helena Pellizzon

**Affiliations:** 1Institute of Biosciences, São Paulo State University (UNESP), Botucatu 18618-689, SP, Brazil; 2School of Pharmaceutical Sciences of Ribeirão Preto, University of São Paulo (USP), Ribeirão Preto 14040-903, SP, Brazil

**Keywords:** antioxidants, skin wound healing, histopathology, Brazilian red propolis, bee products

## Abstract

Skin wound healing is a complex process that requires the mutual work of cellular and molecular agents to promote tissue restoration. In order to improve such a process, especially in cases of impaired healing (e.g., diabetic ulcer, chronic wounds), there is a search for substances with healing properties and low toxicity: two features that some natural products—such as the bee product named propolis—exhibit. Propolis is a resinous substance obtained from plant resins and exudates with antioxidant, anti-inflammatory, and antitumoral activities, among other biological ones. Based on the previously reported healing actions of different types of propolis, the Brazilian red propolis (BRP) was tested for this matter. A skin wound excision model in male Wistar rats was performed using two topical formulations with 1% red propolis as treatments: hydroalcoholic extract and Paste. Macroscopical, histological and immunohistochemical analysis were performed, revealing that red propolis enhanced wound contraction, epithelialization, reduced crust formation, and modulated the distribution of healing associated factors, mainly collagen I, collagen III, MMP-9, TGF-β3 and VEGF. Biochemical analysis with the antioxidants SOD, MPO, GSH and GR showed that propolis acts similarly to the positive control, collagenase, increasing these molecules’ activity. These results suggest that BRP promotes enhanced wound healing by modulating growth factors and antioxidant molecules related to cutaneous wound healing.

## 1. Introduction

Skin wound healing is a highly complex process in which a number of cellular and molecular mediators are mobilized to protect and remove pathogens from the wounded site, creating a new environment where cells must proliferate and finally seal the wound [1]. These events occur in overlapping phases, namely hemostasis, inflammation, proliferation and tissue remodeling, each one having its own pattern of predominant cells, extracellular matrix (ECM) arrangement, cytokines, growth factors, and enzyme activity that orchestrate the entire process [2,3].

The mechanisms to restore skin integrity, when unbalanced, become ineffective, resulting in pathological scenarios such as chronic wounds, diabetic ulcers, psoriasis, pressure ulcers and burn-related wounds—conditions that, in addition to causing pain and morbidity, require a large financial investment in treatments [3,4]. There is also a commercial appeal to skin wound healing treatments, especially those that aim for scarless wounds, avoiding the formation of hypertrophic scars and keloids that are sources of social burden [5,6].

Despite a large number of topical medicines for healing and the development of new drugs–through both synthetic chemistry and biotechnology—natural products continue to be a great source of medicines, some of them having their origin in folk medicine [7]. Medicinal plants and bioactive substances appear as alternative treatments, reducing wound closure time, reepithelialization, and tissue fibrosis due to the presence of secondary metabolites, such as tannins, steroids, terpenes, flavonoids, alkaloids, coumarins, and saponins, that act in the different phases of healing [8].

The antioxidant potential of said bioactive compounds is a noted property, since excessive oxidation, namely oxidative stress, is the cause of delayed healing in several pathological conditions such as diabetes [9,10].

These substances could both act as antioxidant molecules by neutralizing reactive species (e.g., superoxide anion, hydroxyl, and sulfate radicals) and increase the production of antioxidant molecules present in the body. Superoxide dismutase, glutathione and myeloperoxidase are examples of antioxidants produced within the organism, the quantities of which can change due to the presence of molecules from natural compounds [11,12,13].

Similar actions occur between natural compounds and inflammation. The process involves anti- and proinflammatory molecules to stop and stimulate the events, respectively. Some natural compounds have the ability to increase anti-inflammatory agents and decrease proinflammatory agents, which is desirable, especially in cases of delayed wound closure for prolonged inflammation, as seen in chronic wounds [14,15,16].

Propolis, a resinous material collected by bees, mainly *Apis mellifera*, from plant resins and exudates has been studied for its antioxidant, anti-inflammatory, and antitumoral properties, among other biological activities [17]. Its chemical composition relates to secondary metabolites found in the vegetable sources used in propolis production by *Apis mellifera* and other species of bees [18]. Brazilian red propolis is a recently described type of propolis, the main plant source of which is *Dalbergia ecastophyllum (L) Taub.* (Fabaceae), which is abundant in the northeastern region of Brazil [19]. The biological activities of Brazilian red propolis have been widely investigated since propolis preparations are particularly active in antitumor, anti-inflammatory, and immunomodulatory assays, both in vitro and in vivo [17]. A recently published in vitro study by M Afonso et al. (2020) showed that propolis extract promoted fibroblast migration and wound healing, and its anti-inflammatory and antioxidant properties were conferred mainly by the high concentrations of phenolic compounds found in the extract [20].

The plasma bioavailability of phenolic compounds is higher in crude propolis extracts than when the same compounds are applied as isolated molecules, as assessed by Curti et al. (2019) and Boufadi et al. (2017), suggesting that the substances presented in propolis might act in synergistic ways, being advantageous to use the whole product instead of its isolated compounds when regarding its antioxidant activities [21,22].

Therefore, our research focused on investigating the actions of the Brazilian red propolis (BRP) from Bahia, Brazil, when applied to rat skin wounds in two distinct topical formulations (hydroalcoholic extract and paste) at 1% to analyze the natural product potential in improving skin wound healing.

## 2. Materials and Methods

### 2.1. Chemicals and Reagents

Immunohistochemistry kits were purchased from Abcam (Cambridge, UK), and as the primary antibodies against Ki67 (ab16667), VEGF (ab46154), collagen III (ab7778), FGF (ab8880), TGF-β3 (ab15537) and S100A4 (ab41532) were used. α-SMA (CGA7-sc53015) and MMP-9 (C20-sc6840) were purchased from Santa Cruz Biotechnology, Inc., (Dallas, TX, USA) and collagen I (orb312178) was purchased from Biorbyt (Cambridge, UK). The total protein content of samples was determined by the Biuret method (Katal Biotecnológica Ind. Com. Ltd.a., Minas Gerais, Brazil). ELISA kits for TNF-α and IL-10 and the antioxidant enzymes SOD, GSH, GR and MPO were purchased from R&D Systems (Minneapolis, MN, USA). Collagenase 1.2 IU (refuse drug) was purchased from pharmaceutical companies.

### 2.2. Red Propolis Extraction

BRP was obtained in December 2019 from *Apis mellifera* apiaries located in Bahia state in the northeastern region of Brazil (SisGen registration number: AF234D8). After drying and pulverizing, propolis was frozen, ground, and macerated with a 70% ethanol and water solution (7:3, (*v*/*v*)) using a shaker at 30 °C and 120 rpm for 24 h, and the process was repeated until the complete extraction was completed. The extract was then concentrated (vacuum) and lyophilized until the drying was complete, as described by Aldana-Mejía et al. (2021a) [23] and Aldana-Mejía et al. (2021b) [24]. 

### 2.3. Formulation of 1% Red Propolis Hydroalcoholic Extract and Paste

Two formulations containing propolis were developed: hydroalcoholic extract of propolis and paste with propolis, both at 1%, as the best effective concentration defined in the work of Berretta et al. (2018) [25]. The hydroalcoholic extract comprised 50 mL of a 70% ethanol mixture with 500 mg of red propolis. The paste is an established pharmaceutical form classified as a fatty paste for its zinc oxide content. The paste used in this experiment had the following ingredients: 12.5 g of maize starch; 12.5 g of zinc oxide; 12.5 g of calcium dihydroxide; 9.98 mL of glycerin; 500 mg of red propolis; and 0.5 mL of 100% ethanol.

### 2.4. Animals

Male *Wistar* rats were divided into three experimental groups of six animals each (as shown above), and then treated for 3, 7 or 14 days following the evolution of the healing phases, as described in the literature [26]. The study was approved by the Ethics Committee on Animal Use at São Paulo State University under protocol no. 9793211119:Group 1 (WWT/negative control): Wounded animal without treatment;Group 2 (BRP-HS): Wounded animals treated with Brazilian red propolis hydroalcoholic solution at 1%; andGroup 3 (BRP-Paste): Wounded animals treated with paste containing Brazilian red propolis at 1%.

### 2.5. Experimental Protocol of Wound Excision

The animals were subjected to intraperitoneal anesthesia with ketamine (80 mg/kg) and xylazine (10 mg/kg) to apply the wound excision procedure according to the model of Gushiken et al. (2017) [27]. Briefly, the dorsal region of the animals was shaved and submitted to a lesion 2 cm in diameter. Wounds were photographed, measured, and treated with topical formulations daily, according to the respective experimental groups. After the respective periods of treatment (3, 7, and 14 days), the animals were euthanized, and skin wound samples were obtained and stored for histological analysis. 

### 2.6. Macroscopic Analysis

#### 2.6.1. Wound Contraction Analysis

Through photos taken daily by a professional camera—and a ruler for scale—fixed on a stand and kept at a standardized distance, wound retraction was measured using ImageJ (NIH, Bethesda, Maryland, USA) [28], and the wound retraction percentage was calculated as follows [29]: % wound retraction = {(initial area of the wound − final area of wound measured)/initial area of the wound} × 100(1)

#### 2.6.2. Clinical Parameters

Clinical parameters of coagulation, granulation tissue, epithelialization, and the presence of crust were evaluated using a four-point scale classification based on the observation of the photographs of the wounds, as described by de Oliveira et al. (2014) [30]: 0–absence (0%), 1–little (1–30%), 2–fair (30–70%) and 3–very much (>70%), observed in the photographs of the wounds, as described by de Oliveira et al. (2014) [30].

### 2.7. Microscopic Analysis

#### 2.7.1. Histological Parameters

After euthanasia, skin wound samples were processed to make slides for microscopic analysis. Hematoxylin and eosin (HE) dyes, Mallory’s trichrome, and immunohistochemical techniques were used to observe the total content of cells in the dermis and the epidermis (both in square microns) and epidermis thickness (in microns), total content of collagen (in square microns), and specific biological content described in the next section. An optical microscope and Olympus cellSens Software (RRID: SCR_014551) were used to identify the dermal regions of normal (undamaged) skin, borders, and center regions of the wounds (see Figure 1 for these region delimitations), following the methodology of Gushiken et al. (2022) [31].

From every three regions of each sample, 15 images (photomicrographs) were captured and then subjected to planimetric analysis in AvSoftBioView 5©–Spectra software, which calculates the sum of areas delimited by the experimenter generating values in square microns that were used in the statistical calculations (considering a total area of 100,000 μm^2^/slice). This protocol is based on the work of Gushiken et al. (2022) [31].

#### 2.7.2. Immunohistochemistry

Skin samples were subjected to standard processes for immunohistochemistry techniques. Briefly, the samples were fixed with 10% buffered formaldehyde, embedded in paraffin, and sectioned to obtain 5 μm thick sections. The slices were then submitted to antigen retrieval by pressure (20 psi/125 °C). Immunohistochemistry was performed with antibodies against α-SMA (1:100 µL), collagen I (1:200 µL), collagen III (1:100 µL), FGF (1:10,000 µL), Ki67 (1:100 µL), MMP-9 (1:100 µL), S100A4 (1:100 µL), TGF-β3 (1:100 µL) and VEGF (1:100 µL).

For VEGF immunostaining, two different analyses were performed: the quantification of blood vessels by a hand counter, considering the vessel morphology in the dermis of each sample; and the quantification of the area in square microns containing cells immunostained with the anti-VEGF antibody using AvSoftBioView 5©–Spectra software as described in the last section [31]. 

### 2.8. Antioxidant Enzyme and Inflammatory Mediator Analysis

For the analysis of GSH, GR, MPO, and SOD enzymes and inflammatory mediators, we used a scalpel to obtain samples of half of the wounds, a piece that included both border and central regions. These samples were homogenized (1:5 *m*/*v*) in phosphate buffer (pH 7.4), and after centrifugation (5 min, 10,000 rpm, 4 °C), the supernatant was collected. To assess the total protein content of the samples, we performed the biuret method for protein quantification according to the instructions given in the kit. The anti-inflammatory cytokine IL-10 and the proinflammatory cytokine TNF-α were quantified following the instructions provided by the supplier, and the results are expressed as pictograms per milliliter (pg/mL). Biochemical assays for the analysis of GSH [32], GR [33] MPO [34], and SOD [34], based on specific protocols, generated results expressed as nanomoles per microgram (nmol/mg) of protein, units per gram (U/g) of protein, and units per milligram (U/mg) of protein, respectively.

### 2.9. Statistical Analysis

Clinical parameters are expressed as the median and analyzed by Kruskal-Wallis test with Dunn’s posttest. Histological parameters are expressed as the mean ± standard error of the mean. Wound contraction is expressed as the mean ± standard deviation. Comparisons between groups were made by one-way ANOVA followed by the Newman-Keuls test. The analyses were performed using GraphPad Prism software, version 8.0.1 (GraphPad Software Inc., San Diego, CA, USA) with 5% significance.

## 3. Results

### 3.1. Macroscopic Analysis

#### 3.1.1. Wound Contraction Analysis

Through measurements made during the periods of 3, 7, and 14 days, wound contraction percentages were analyzed and the results are shown in Figure 2. Propolis treatments did not cause significantly different results in wound contraction compared to the WWT group.

#### 3.1.2. Clinical Parameters

The parameters of coagulation, granulation tissue, epithelialization, and crust presence were analyzed macroscopically, and the data were classified into a four-point scale: 0–absence (0%), 1–a little (1–30%), 2–fair (30–70%) and 3–very much (>70%). We found a significant difference only in the groups treated for 14 days. Thus, Figure 3 presents data from the 14 day treatment groups. Both propolis-containing formulations (BRP-HS and BRP-Paste) showed less crust formation than the WWT group. The BRP-HS group also showed a better epithelialization process than the WWT group.

### 3.2. Microscopic Analysis

#### 3.2.1. Histological Parameters

Figure 4 shows some of the 15 fields of analysis for HE and Mallory’s trichrome stained samples. No significant difference regarding total cell count in the dermis was obtained, as shown in Appendix A. Total collagen qualitative analysis showed no statistical significance in any of the treated groups, except for the BRP-Paste group at the border of the wounds, which had increased collagen immunolabeling compared to WWT at 14 days (Appendix A). 

Regarding the epidermal thickness throughout the treatment periods, the BRP-paste treatment kept the epidermis thinner than in the WWT group at 7 and 14 days of treatment, as shown in Figure 5.

#### 3.2.2. Immunohistochemistry

Photomicrographs of samples submitted to immunohistochemistry can be found in the Appendix A.

α-SMA

The immunohistochemical results for α-SMA showed decreased labeling in both propolis-containing formulations after 3 days of treatment at the center of the wound. After 7 days, BRP-HS and, more intensively, BRP-Paste had less labeling for α-SMA at the border and center of the wounds. Similar results were observed at 14 days of treatment compared to WWT (Appendix A).

Collagen I

Immunohistochemistry for collagen I showed increased labeling in all groups at the border of the wounds after 3 days of treatment, with the BRP-HS and BRP-Paste groups having similar significance. At 14 days, still at the border, BRP-Paste presented less labeling for collagen I. At the center of the wounds, no significant difference was observed at 3 days compared with 7 days of treatment. Groups BRP-HS and BPR-Paste showed less labeling than WWT, and at 14 days, fewer collagen I-positive areas were observed in the BRP-Paste group (Figure 6 and Appendix A).

Collagen III

Immunohistochemistry results for collagen III showed fewer labeled areas at the border of the wounds in 3 days for the BRP-HS group. At 7 days, both propolis treatments showed more labeled areas for collagen III at the center of the wounds. At 14 days, however, BRP-HS presented more collagen III-positive areas than WWT, while BRP-Paste followed the same pattern of decreased collagen III expressions as observed at 7 days (Figure 7 and Appendix A).

FGF

Immunohistochemistry of FGF showed that, at the border of the wounds, BRP-Paste treatment for 3 days resulted in decreased labeling. At the center of the wounds, during the same period, all treatments led to fewer FGF-positive areas. No statistical significance was observed at 7 days in any region, while at 14 days of treatment, at the center of the wounds, BRP-HS and BRP-Paste samples showed fewer labeled areas for FGF, with the latter more intensively, as shown in Appendix A.

Ki67

Immunohistochemical results of Ki67 showed fewer labeled areas in the BRP-HS and BRP-pPaste group at the center of the wounds at 3 and 7 days of treatment. There were no significant differences in the results for border regions and the 14-day treatment period, as presented in Appendix A.

MMP-9

Immunohistochemistry of MMP-9 showed less labeling in the BRP-HS and BRP-Paste groups at the center of the wounds at 3, 7, and 14 days of treatment. At 7 and 14 days, the border region also presented fewer MMP-9 areas in the propolis-treated groups than in the WWT group, as shown in Figure 8 and Appendix A.

S100A4

Immunohistochemical results for S100A4 at the border of the wounds after 7 days of treatment showed decreased labeling in the BRP-HS group and a more intense decrease in the BRP-Paste group. At 14 days at the border of the wounds, the BRP-HS-treated groups presented more S100A4-labeled areas than WWT, while BRP-Paste resulted in decreased labeling for S100A4 (Appendix A).

TGF-β3

Immunohistochemistry for TGF-β3 showed less labeling at 7 days of treatment at the border region for the BRP-HS- and BRP-Paste-treated groups. At the center of the wounds, only BRP-Paste presented less labeling for TGF-β3. At 14 days at the border, both propolis treatments led to decreased TGF-β3-positive areas, as shown in Figure 9 and Appendix A.

VEGF

Immunohistochemical results—analyzed by AvSoftBioView 5© software, which quantifies the area in µm^2^—for VEGF showed increased labeling at 3 and 7 days of treatment at the border and center of the wounds in the BRP-HS and BRP-Paste groups compared to the WWT group (Figure 10 and Figure 11).

In the hand counter quantification of VEGF, a similar pattern of results was obtained: after 3 days of treatment: the propolis-treated groups showed higher quantities of blood vessels at the border. After 7 and 14 days of treatment, however, no significant differences were observed in the hand counter quantification of VEGF, as shown in Figure 12.

### 3.3. Antioxidant Molecule and Inflammatory Mediator Quantification

TNF-α and IL-10 quantities measured by the ELISA technique, presented in Figure 13, were not significantly different among any of the compared groups (Col and BRP-HS were all treated for 14 days). 

Regarding the antioxidant activity of the SOD, GR and MPO enzymes, BRP-HS-treated samples presented no difference in the activity of these molecules compared to group COL. For GSH quantification, the BRP-HS-treated groups presented more enzyme activity, as shown in Figure 14.

## 4. Discussion

Brazilian red propolis is one of the 14 types of propolis found in the country [30]. The red propolis samples used in the present study were gathered from beehives in Bahia, a state located in the northeastern region of Brazil. *Dalbergia ecastaphyllum (L.) Taub*. (Fabaceae) is the main botanical source of most bioactive compounds, as previously confirmed by other studies using red propolis from the same location [23,35,36]. Through chemical characterization and quantification of red propolis compounds made by Aldana-Mejía et al. (2021a) [23] and Aldana-Mejía et al. (2021b) [24], three major molecules were identified: vestitol, medicarpin, and neovestitol. All of them are isoflavonoids found both in propolis and in *Dalbergia ecastaphyllum* branches [37]. Isoflavonoid derivatives are also found in high concentrations in the Fabaceae genera for their defensive properties to these plants;. These molecules could show antimicrobial, antioxidant, and anti-inflammatory effects–related to their function on plants–when applied to other living organisms [38]. Based on these facts, the Brazilian red propolis was investigated for its actions on skin wound healing.

The first parameters to be analyzed were macroscopic, in which wound contraction and clinical parameters were evaluated. Wound contraction is an event motivated by the mechanical forces of contractile cells, such as myofibroblasts, and extracellular matrix fibers, such as fibronectin and collagen, leading to wound closure [39,40]. Macroscopic observations of the wounds revealed no differences in the percentage of wound closure among the propolis-treated groups and the nontreated group. However, the wounds showed better epithelialization processes in the BRP-HS group, as well as less crust formation when treated with BRP-HS and BRP-Paste compared to untreated animals after 14 days of treatment. This outcome is an indication that propolis can more actively influence the final stages of wound healing, as confirmed by other analyses. 

The vehicles—hydroalcoholic solution and paste—used in this experiment did not influence propolis’ actions on skin, as presented in previous studies and by comparison between animals treated only with vehicle and those that received vehicle and propolis (not shown) [41,42]. In particular, the paste promoted good adhesion of the drug or bioactive product in question, with special drying effects for the absorptive feature of the powder containing it, providing an adequate medium for the performance of the drug [43].

The macroscopic aspects of wound healing are influenced by microscopic events that occur in the skin. Thus, our research focused on factors associated with the extracellular matrix (ECM) constitution (collagen I and III; MMP-9) and with cell proliferation, migration, and differentiation (FGF; S100A4; Ki67; TGF-β3) by immunohistochemical qualitative analysis.

MMP-9 is an endopeptidase with an important role in degrading ECM components, allowing for cell migration and new vessel formation (angiogenesis) [44]. Both propolis formulations decreased MMP-9 immunolabeling after 3, 7, and 14 days of treatment, which could be interpreted as a control on the activity of the enzyme to avoid excessive degradation of the tissue being formed. Most of the ECM during the proliferative phase of wound healing consists of collagen III fibers—with mechanical properties that facilitate the massive traffic of cells that occurs during the proliferative phase—which are gradually substituted by collagen I fibers, which, due to their characteristics, render the tissue more intact and stable as the re-epithelialization phase begins [45]. Our microscopic analysis of total collagen, collagen I, and collagen III distribution showed that BRP-Paste increased collagen I at 3 days, increased total collagen at 14 days, decreased collagen III at 7 days, and decreased collagen I at 14 days of treatment. BRP-HS increased collagen I at 3 days, decreased the same type of collagen at 7 days, and decreased collagen III at 3 and 7 days, but at 14 days, it increased the positive area for collagen III, which is relatable to the physiologic pattern of collagen distribution in skin wound healing. This action also corroborates the assumption that propolis might favor scarless healing by increasing the presence of collagen III [46,47]. Thus, Brazilian red propolis influences collagen fiber distribution through the wound healing phases, improving its completion.

The abnormal deposition of extracellular matrix components causes an undesired situation called fibrosis that alters the architecture of the tissue, compromising its functioning [48]. Fibroblasts are connective tissue cells with an important role in fibrosis since they produce components of the ECM during wound healing [49]. 

Among the factors with antifibrosis capabilities is basic FGF (fibroblast growth factor, FGF-2 or bFGF), a hormone that stimulates proliferation and migration during wound healing, also diminishing the production of collagen I fibers and fibroblast differentiation into myofibroblasts [50,51]. bFGF was decreased at 3 and 14 days of treatment by both propolis formulations in our experiment, suggesting a profibrotic action of propolis.

Other molecules that act toward healing are the S100 proteins, which are found in the ECM and in cells, influencing motility, stimulating cytoskeletal rearrangement, cell proliferation, ECM remodeling, collagen I production and angiogenesis [52]. Our results showed that treatment with propolis in an extract significantly increased S100A4 (one isoform of S100 proteins) at 7 days and decreased it at 14 days of treatment. Conversely, BRP-Paste only decreased S100A4-positive areas in both treatment periods. S100A4 plays a role in inducing fibrosis, stimulating the release of proinflammatory factors and MMPs (including MMP-9), and it is involved primarily in remodeling processes [53]. BRP-HS once more seems to act as an adequate propolis formulation by both improving collagen distribution and diminishing S100A4 presence, thus avoiding fibrosis.

To confirm whether propolis influences cell proliferation, we analyzed another protein called Ki67, which is expressed by cells that are undergoing the G1, G2, S, and M cell cycle phases (corresponding to proliferative events) by immunohistochemistry [54]. Our results showed decreased labeling of Ki67 in the dermis after 3 and 7 days of treatment with both propolis formulations. This outcome corroborates the analysis of epidermal thickness: propolis treatments decreased the length of the epidermis when compared to the WWT group. This finding indicates that cell proliferation was not exacerbated, and it occurs in the formation of abnormal scars, such as keloids and hypertrophic scars [55].

The last phases of wound healing are marked by the presence of contractile cells called myofibroblasts, which are specialized fibroblasts with stress fibers known as α-SMA fibers in their cytoplasm that assist in mechanical wound contraction [56]. Excessive myofibroblast presence relates to excessive scarring. Recent research has pointed out that keloids are characterized by a large number of α-SMA-positive cells in the dermis [57]. Since propolis treatments led to fewer α-SMA-labeled areas at 3, 7 and 14 days of treatment, there is a chance that the natural product could act as an antiscarring agent, a statement that requires confirmation by further analysis.

Excessive scarring is an undesired condition that seems to be attenuated by the presence of a growth factor known as TGF-β3 according to recent studies [58,59]. TGF-β3 was found to play an important role in most wound healing phases (especially in the inflammation phase) by promoting reduced ECM synthesis and increased MMP activity, all of which culminated in a scar-reducing effect [57,58]. Propolis treatment resulted in lower TGF-β3-positive areas after 7 and 14 days of treatment, which could be interpreted as a pro-scarring action, as was also corroborated by the low levels of MMP-9 after 14 days of treatment. However, TGF-β3 has pleiotropic qualities that are not completely understood, and since the macroscopic data concern only wound closure (and not the scarring aspect afterward), not much was concluded from these results [60]. 

Finally, Brazilian red propolis treatment for 3, 7, and 14 days resulted in increased VEGF immunolabeling. VEGF stimulates angiogenesis during the proliferative phase of wound healing by its mitogenic and chemotactic actions, providing oxygen and nutrient access to the new cells at the wound site [61]. In addition to being produced by endothelial cells, VEGF is found in macrophages, keratinocytes, and myofibroblasts [62,63]. Thus, the qualitative VEGF count performed in the dermis region also considered the hormone’s presence in macrophages and myofibroblasts beyond the endothelium. To confirm BRP actions toward the VEGF and angiogenesis results, the number of blood vessels was counted, and propolis-treated groups showed more vessels than WWT groups at 3 days of treatment as well. Therefore, red propolis might act as a proangiogenic substance. 

By analyzing significant results from immunohistochemistry at the border and center regions of the wounds in all groups, we concluded that the main parameters that were possibly altered by the action of Brazilian red propolis were collagens I and III, MMP-9, TGF-β3 and VEGF. In particular, treatment with propolis hydroalcoholic extract resulted in more significant alterations throughout the three different periods of treatment than the paste formulation compared to WWT. BRP-Paste showed different actions from BRP-HS in some of the parameters, such as in collagen I (it had no effect over 14 days, while BRP-HS decreased collagen I labeling), collagen III (in which BRP-Paste decreased collagen III and BRP-HS increased it in 14 days of treatment), and S100A4 in 14 days, BRP-Paste decreased the presence of S100A4, while BRP-HS increased it. However, it is still not possible to conclude that one formulation has higher wound healing capacity than another.

As assessed by previous studies, BRP presents antioxidant actions that can prevent oxidative stress and related mechanisms that delay healing [17,64,65,66,67]. To some extent, the presence of oxidative species, such as reactive oxygen species (ROS), primarily in the inflammatory phase of wound healing, is necessary for the elimination of invading pathogens, as well as functioning as signaling molecules to recruit lymphocytes to stimulate vasoconstriction, among other assignments [68]. However, if they occur in great amounts and for long periods, ROS can impair wound healing, as observed in chronic wounds, which is why antioxidant enzymes are important [69].

The biochemical quantification of antioxidant molecules and cytokines was performed after the previously described macro- and microscopic analyses and included a positive control group, the skin wounds which were treated with collagenase 1.2 IU (COL group)—a standard formulation for skin injuries—following the methodology of Gushiken et al. (2022) [31]. All groups were treated for 14 days considering previous data from Gushiken et al. (2022) [31], who emphasized with their results that this period of treatment is adequate for the evaluation of wound healing properties when collagenase is used. Since the macroscopic evaluation of wound contraction and clinical parameters in BRP-Paste-treated animals was prejudiced by the whitish aspect of the paste, and the results were better described in the 14-day treated groups, only BRP-HS-treated samples were used for the biochemical analysis. 

Superoxide dismutase (SOD) and myeloperoxidase (MPO) had similar activities in both BRP-HS and COL, suggesting that propolis exerts the same antioxidant stimulus as collagenase, a commonly used skin wound healing drug. SOD was recently reported to play a role in suppressing inflammation in skin wounds, highlighting the importance of this enzyme’s activity [70]. MPO, in contrast, is already linked to the control of innate immune cell activation, thus regulating the inflammatory phase in wound healing [71]. Glutathione (GSH) and glutathione reductase (GR) are also antioxidant molecules present in wound sites responsible for the protection of oxidation-vulnerable regions of important proteins [72]. The results showed that skin wound samples treated with Brazilian red propolis had higher GSH concentrations than those treated with collagenase. Other experiments have presented propolis as a GSH-increasing substance in the wound region, confirming its antioxidant action [73,74]. 

The concentrations of the anti-inflammatory and proinflammatory cytokines IL-10 and TNF-α were also evaluated based on previous reports about the ability of BRP to increase proinflammatory factors and decrease anti-inflammatory factors [75,76]. No significant differences in IL-10 or TNF-α levels between collagenase and propolis treatments were observed, suggesting that propolis might act as an anti-inflammatory by maintaining adequate levels of both cytokines, such as collagenase.

The fact that the macroscopic visualization of wound contraction and clinical parameters were hampered by the whitish aspect of the paste might be the reason why the BRP-Paste group did not show more significant results in a general analysis, mainly at 3 and 7 days of treatment. Nonetheless, Brazilian red propolis, both in an extract and in a paste, seems to exert beneficial effects on skin wound healing, as revealed by macroscopic results in epithelialization and crust formation and by microscopic analysis of the presence of healing-associated factors, such as collagen I and III, MMP-9, TGF-β3, and VEGF. The natural product acts by enhancing the activity of the antioxidant enzymes SOD and MPO, modulating GR and GSH and controlling the presence of anti-inflammatory (IL-10) and proinflammatory (TNF-α) cytokines. 

BRP has a multitude of future applications as an antioxidant, antimicrobial, antitumoral and skin wound healing enhancer substance. From the perspective of our study, there is a need for molecular investigations on skin wound healing related molecules associated with further in vitro tests that would expand the knowledge on propolis action mechanisms. Since propolis already has potential future applications in different fields–such as in dentistry for its antimicrobial action [77]—the next steps are to understand how all the compounds that form propolis interact with each other to produce these effects, thus it will be possible to achieve optimal formulations for each clinical condition. 

## 5. Conclusions

Considering the results, Brazilian red propolis added to a hydroalcoholic solution and into a paste, both at 1%, has beneficial effects on wound healing by modulating the expression of several molecules related primarily to the proliferative and tissue remodeling phases, evidenced by macroscopic and microscopic analysis. With regard to the epithelialization process, the presence of TGF-β3 as well as other cell proliferation-related proteins (Ki67, S100A4, and FGF) showed that propolis acts in favor of wound closure by maintaining these molecules in adequate distribution throughout the wound healing phases. Propolis also modifies the constitution of the newly formed ECM by modulating MMP-9 actions and the angiogenesis process (VEGF), as well as the epidermal layer, keeping it at normal ranges. The substance regulates the activity of antioxidant enzymes (SOD, MPO, GSH and GR), avoiding any oxidative stress that would prevent healing. Therefore, the present study demonstrated that Brazilian red propolis influences physiological skin wound healing and that the natural product has suitable properties that can be used in this regard.

## Figures and Tables

**Figure 1 antioxidants-11-02188-f001:**
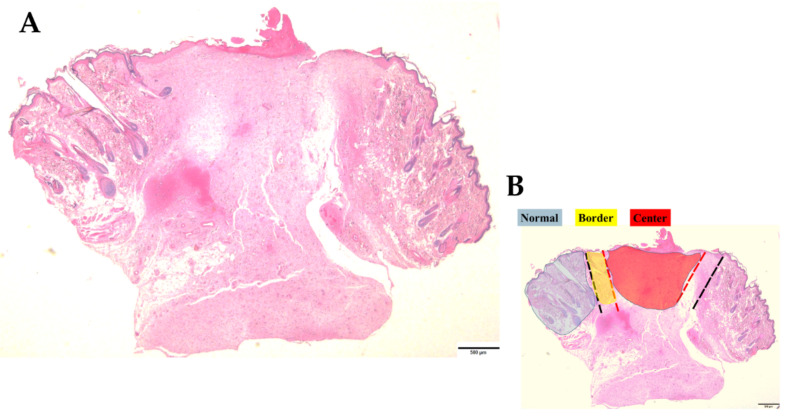
Schematic image of a skin sample stained with Hematoxylin and Eosin (HE). (**A**) shows a whole vision of a skin sample by a microscope in 2× lens. (**B**) schematizes the three analyzed regions in the experiment (unwounded or normal skin, border and center of the wounds), delimitating them by colors: the normal region in blue, the border in yellow and the center in red. Bars 500 µm.

**Figure 2 antioxidants-11-02188-f002:**
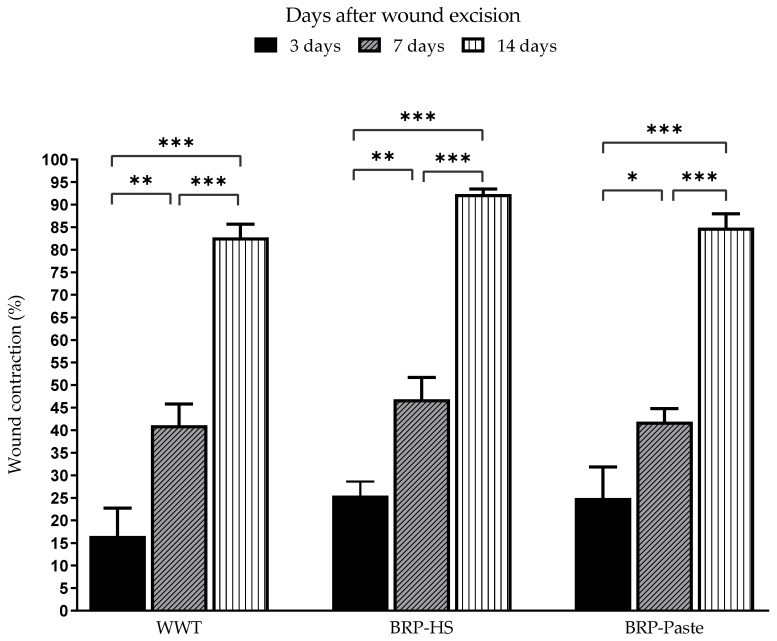
Wound contraction percentage of surgical skin wounds in Wistar rats treated over 3, 7 and 14 days. * *p*< 0.05, ** *p* < 0.01 and *** *p* < 0.001 in comparison to the WWT group by one-way analysis of variance (ANOVA) followed by Newman-Keuls test (*n* = 6).

**Figure 3 antioxidants-11-02188-f003:**
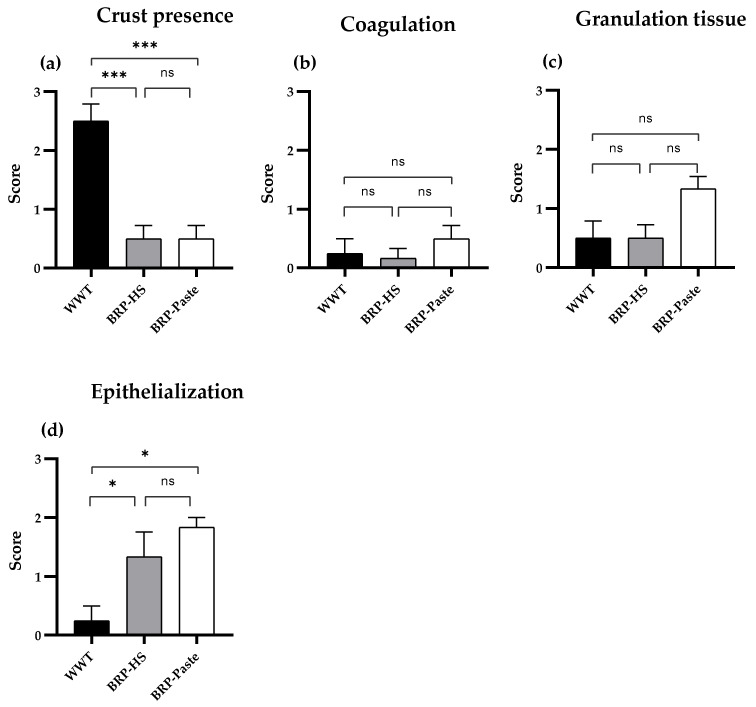
Clinical parameters appearance score of skin wounds after 14 days of treatment. Data shows macroscopical and qualitative evaluation of the clinical aspects of crust presence (**a**), coagulation (**b**), granulation tissue (**c**) and epithelialization (**d**) in which 0–absence (0%), 1–a little (30%), 2–fair (30–70%) e 3–very much (>70). Statistical analysis by Kruskal-Wallis test followed by Dunn’s test. * *p* < 0.05, *** *p* < 0.001 and ns = no significance compared to the WWT group (*n* = 6). WWT: wounded animals without treatment; BRP-HS: hydroalcoholic solution with 1% Brazilian red propolis treatment; BRP-Paste: paste with 1% red propolis treatment.

**Figure 4 antioxidants-11-02188-f004:**
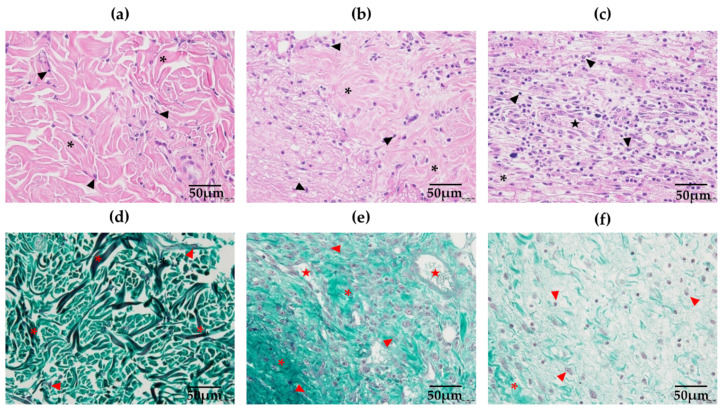
Photomicrographs in 40× lens of the normal (unwounded region) (**a**), border (**b**) and center (**c**) regions of the wounds at the dermis in HE (**a**–**c**) and Mallory’s trichrome (**d**–**f**) stained samples. Arrowheads indicate the nucleus of cells that are abundant at the center of the wounds, which also presents a higher number of blood vessels indicated by a star; collagen fibers surrounding the cells are indicated by the asterisk, being well organized in the unwounded region of the skin and becoming more disorganized at the center of the wound. In the Mallory’s trichrome stained samples, abundant and organized collagen fibers (indicated as asterisks) are found at the unwounded region, being replaced by disorganized and shred fibers at the border and center. Bars 50 µm.

**Figure 5 antioxidants-11-02188-f005:**
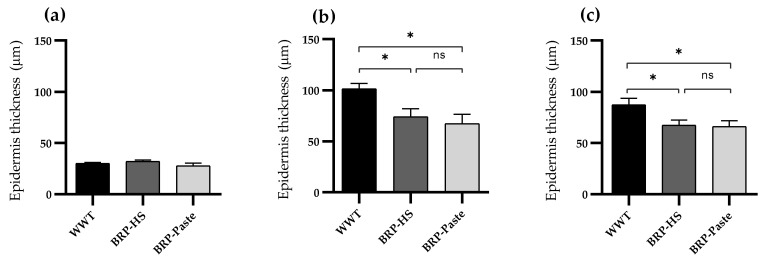
Epidermis’ thickness (μm) of groups WWT, HS, BRP-HS, Paste and BRP-Paste in 3 (**a**), 7 (**b**) and 14 days (**c**) of treatment, analyzed in HE stained skin samples. Results expressed according to one-way ANOVA followed by Newman-Keuls test, * *p* < 0.05 (*n* = 6). ns = no significance.

**Figure 6 antioxidants-11-02188-f006:**
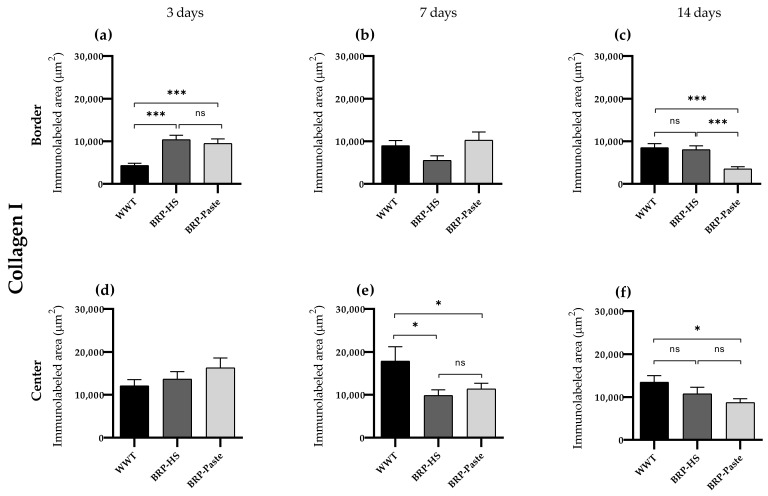
Quantification of collagen type I by immunolabed area (μm^2^) at border (**a**–**c**) and central (**d**–**f**) regions (dermis) of the wounds in each experimental group for the periods of 3 (**a**,**d**), 7 (**b**,**e**) and 14 (**c**,**f**) days of treatment. The statistical difference compared to the WWT group was obtained by one-way ANOVA followed by a Newman-Keuls test in which * *p* < 0.05 and *** *p* < 0.001 (*n* = 6). ns = no significance.

**Figure 7 antioxidants-11-02188-f007:**
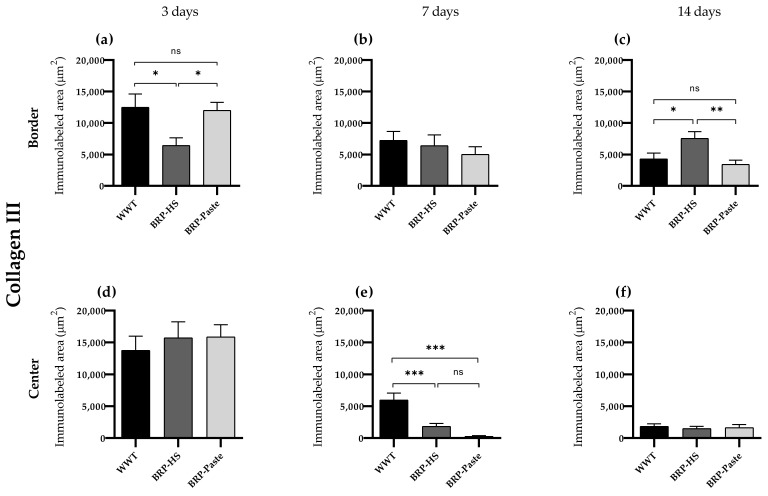
Quantification of collagen type III by immunolabed area (μm^2^) at border (**a**–**c**) and central (**d**–**f**) regions (dermis) of the wounds in each experimental group for the periods of 3 (**a**,**d**), 7 (**b**,**e**) and 14 (**c**,**f**) days of treatment. Statistical difference compared to the WWT group was obtained by one-way ANOVA followed by Newman-Keuls test in which * *p* < 0.05, ** *p* < 0.01, and *** *p* < 0.001 (*n* = 6). ns = no significance.

**Figure 8 antioxidants-11-02188-f008:**
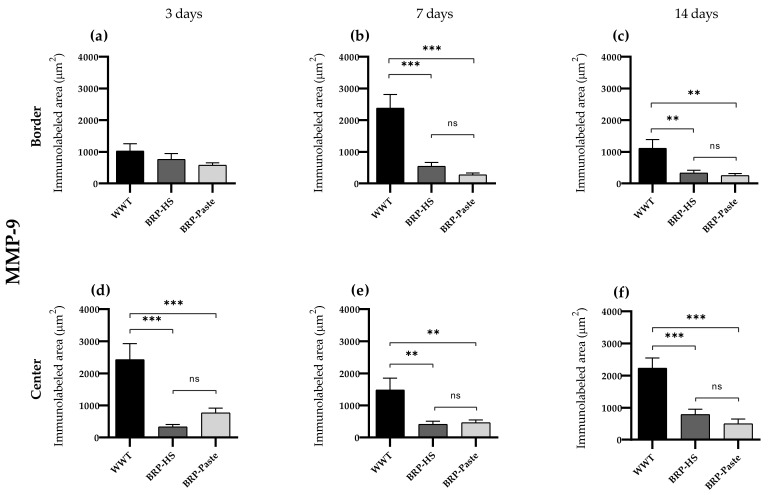
Quantification of the MMP-9 by immunolabed area (μm^2^) at border (**a**–**c**) and central (**d**–**f**) regions (dermis) of the wounds in each experimental group for the periods of 3 (**a**,**d**), 7 (**b**,**e**) and 14 (**c**,**f**) days of treatment. Statistical difference compared to the WWT group was obtained by one-way ANOVA followed by Newman-Keuls test in which ** *p* < 0.01, and *** *p* < 0.001 (*n* = 6). ns = no significance.

**Figure 9 antioxidants-11-02188-f009:**
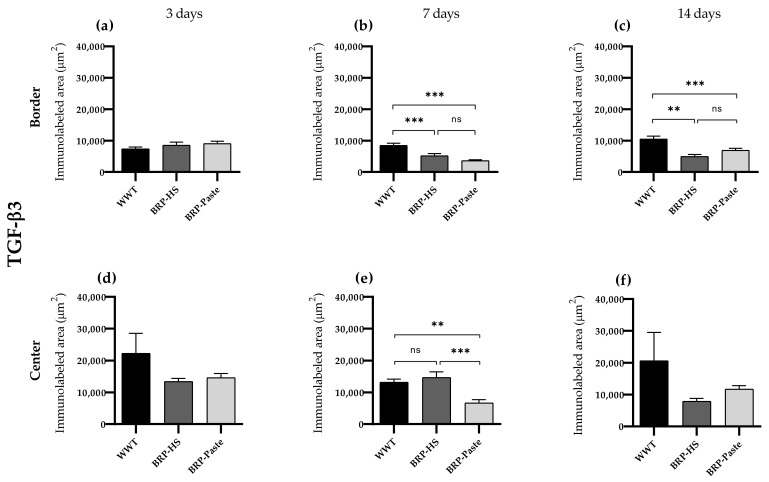
Quantification of TGF-β3 by area (μm^2^) at border (**a**–**c**) and central (**d**–**f**) regions (dermis) of the wounds in each experimental group for the periods of 3 (**a**,**d**), 7 (**b**,**e**) and 14 (**c**,**f**) days of treatment. Statistical difference compared to the WWT group was obtained by one-way ANOVA followed by Newman-Keuls test in which ** *p* < 0.01, and *** *p* < 0.001 (*n* = 6). ns = no significance.

**Figure 10 antioxidants-11-02188-f010:**
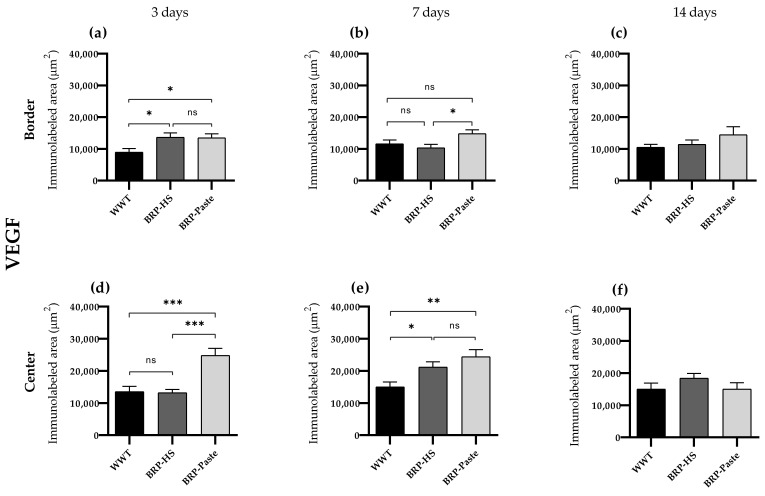
Quantification of VEGF by area (μm^2^) at border (**a**–**c**) and central (**d**–**f**) regions (dermis) of the wounds in each experimental group for the periods of 3 (**a**,**d**), 7 (**b**,**e**) and 14 (**c**,**f**) days of treatment. The statistical difference compared to the WWT group was obtained by one-way ANOVA followed by a Newman-Keuls test in which * *p* < 0.05, ** *p* < 0.01, and *** *p* < 0.001 (*n* = 6). ns = no significance.

**Figure 11 antioxidants-11-02188-f011:**
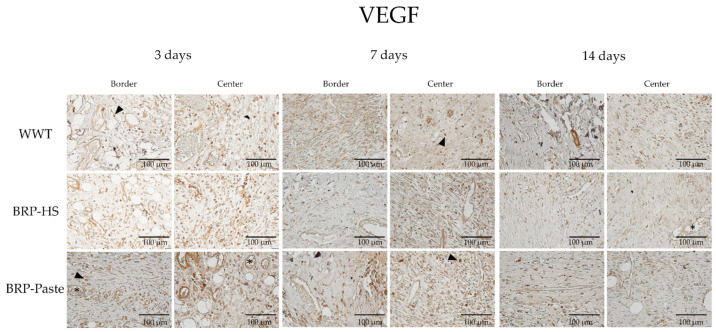
Photomicrographs of the dermis (40×), border and central regions of the wound from each experimental group in 3, 7 and 14 days of treatment for VEGF immunolabeling. VEGF positive cells in brown are represented by arrowheads and blood vessels by asterisks.

**Figure 12 antioxidants-11-02188-f012:**
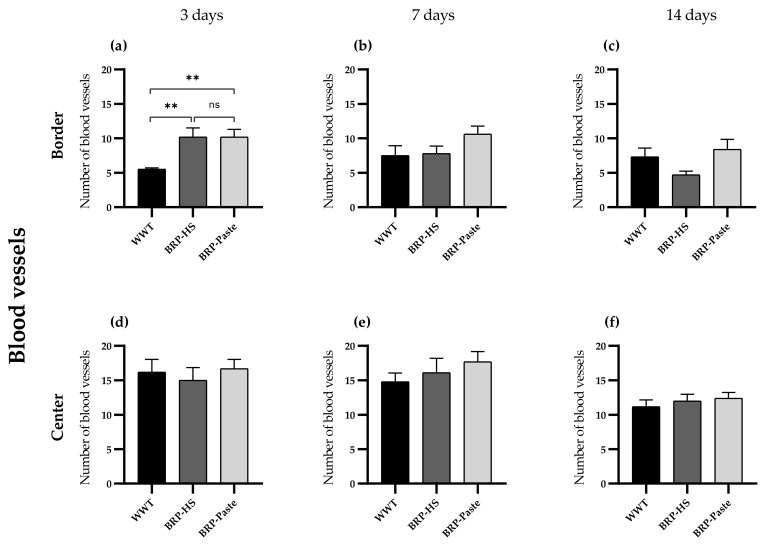
Number of blood vessels in the border (**a**–**c**) and center (**d**–**f**) of the wounds (dermis) in groups WWT, HS, BRP-HS, Paste and BRP-Paste after 3 (**a**,**d**), 7 (**b**,**e**) and 14 (**c**,**f**) days of treatment. Statistical difference according the Kruskall-Wallis test followed by Dunn’s posttest (*n* = 6) in which ** *p* < 0.01. ns = no significance.

**Figure 13 antioxidants-11-02188-f013:**
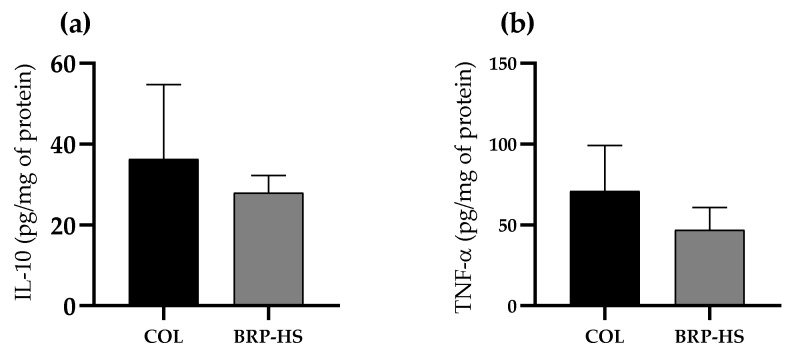
IL-10 (**a**) and TNF-α (**b**) quantification results in pg/mg of protein in groups treated over 14 days with Collagenase 1.2 UI (COL) or BRP-HS. No significance was found according to an unpaired *t* test (*n* = 6).

**Figure 14 antioxidants-11-02188-f014:**
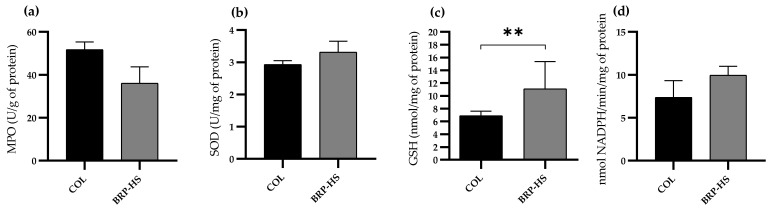
Concentration of MPO (**a**), SOD (**b**), GSH (**c**) and GR (**d**) on skin wound samples from groups COL and BRP-HS treated over 14 days. Statistical significance between groups according to unpaired t test with ** *p* < 0.01 (*n* = 6). No significance between groups in MPO, SOD and GR analysis.

## Data Availability

Data are contained within the article and in the Appendix A link.

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
