# Peer review of "Histological, Immunohistochemical and Antioxidant Analysis of Skin Wound Healing Influenced by the Topical Application of Brazilian Red Propolis"

_antioxidants, 2022, doi:10.3390/antiox11112188_

Round 1

Reviewer 1 Report

The manuscript “Histological, immunohistochemical and antioxidant analysis of skin wound healing influenced by the topical application of Brazilian red propolis” by Mariana Conceição et al. They have; reported that Brazilian red propolis has an important role as wound healing properties. The manuscript is written in standard English with several grammatical errors. I have several comments that needs to be addressed.

Comments:

1.     As authors have focused on antioxidant pathways in the current manuscript and measured the levels of different oxidative stress-related markers. They have measured endogenous non-enzymatic antioxidants such as glutathione (GSH). I will suggest that authors can add GSH-dependent enzymes (GR or GPx or GST) to the current manuscript, which can improve the quality of the manuscript.

2.     I will suggest adding a graphical abstract, which is the point of attraction for a reader.

3.     The authors should check grammatical errors very carefully before resubmission.

4.     In section 2.4. Animals, authors haven’t mentioned how the wounds were formed.

5.     Magnification of IHC images is missing.  

6.     How were the doses of red propolis selected?

Reviewer 2 Report

Dear authors,

After the review process, I have several comments: the abstract is not an introduction, and it should be rewritten; each section in Materials and Methods should have references; in figure 2 not all statistical data are included; the format of the data from table 1 is not correct; in the introduction, you should add new findings about the bioactive potential of functional products and bioavailability of phenolic compounds; in section 4, you should present as the future application of the study the bioactive compounds from natural products, with separation, applications etc.

Best regards.

Round 2

Reviewer 2 Report

Dear authors,

No other comments compared with the first review.

Best regards!